# The Role of Ranolazine for the Treatment of Residual Angina beyond the Percutaneous Coronary Revascularization

**DOI:** 10.3390/jcm9072110

**Published:** 2020-07-04

**Authors:** Simone Calcagno, Fabio Infusino, Nicolò Salvi, Temistocle Taccheri, Riccardo Colantonio, Emanuele Bruno, Lucia Ilaria Birtolo, Paolo Severino, Carlo Lavalle, Mariateresa Pucci, Gennaro Sardella, Massimo Mancone, Francesco Fedele

**Affiliations:** Department of Clinical, Internal, Anesthesiology and Cardiovascular Sciences, Umberto I Hospital, Sapienza University of Rome, 00161 Rome, Italy; calcagnosimone@libero.it (S.C.); fabio.infu@gmail.com (F.I.); nicolo.salvi@hotmail.it (N.S.); taccheri.temistocle@gmail.com (T.T.); riccardo.colantonio@gmail.com (R.C.); emanuele.bruno@email.it (E.B.); ilariabirtolo@gmail.com (L.I.B.); paolo.severino@uniroma1.it (P.S.); carlo.lavalle@uniroma1.it (C.L.); puccimariateresa@gmail.com (M.P.); rino.sardella@uniroma1.it (G.S.); francesco.fedele@uniroma1.it (F.F.)

**Keywords:** percutaneous coronary intervention, complete revascularization, multivessel disease, stable coronary disease, angina, ECG stress test

## Abstract

Introduction. Despite a successful percutaneous coronary intervention (PCI), several studies reported that the recurrence of angina after revascularization, even complete, is a particularly frequent occurrence in the first year after PCI. Methods. The aim was to evaluate the efficacy of treatment with ranolazine in addition to conventional anti-ischemic therapy in patients who underwent coronary angiography for persistent/recurrent angina after PCI and residual ischemia only due to small branches not suitable for further revascularization. Forty-nine consecutive patients were included in our registry, adding the ranolazine (375 mg b.i.d) to optimal medical therapy (OMT). The Exercise ECG Test (EET) was performed in all patients before to start the therapy (baseline BL) and at 30 days (T1) after enrollment. Results. The average duration of the exercise was increased after the therapy with ranolazine comparing to baseline (RG 9’1” ± 2’ versus BL 8’10” ± 2’, *p* = 0.01). Seven (14.3%) patients after receiving ranolazine had not crossed the threshold of six minutes (75 watts) compared to 20 (40.8%) of BL (*p* = 0.0003). Stress angina appeared more frequently at BL than at 30 days (T1 4.1% versus BL 16.3%, *p* = 0.04) as well as exercise-induced arrhythmias (BL 30.6% versus T1 14.3%, *p* = 0.05). Conclusions. The addition of ranolazine to standard anti-ischemic therapy showed a significant improvement in EET results after one month of therapy, including reduced exercise angina, increased exercise tolerance, and reduced exercise arrhythmias.

## 1. Introduction

Stable angina is a chronic medical condition that is generally regarded as the main symptomatic manifestation of chronic coronary syndrome due to a temporary imbalance between myocardial perfusion and metabolic demand causing mainly by epicardial vessel stenosis [1]. The administration of antianginal drugs is the first-line treatment in symptomatic patients, but if symptoms are not satisfactorily controlled, the percutaneous coronary revascularization is recommended [2]. Despite a efficacious percutaneous coronary intervention (PCI) and the long-term procedural success of new stent generations [3,4], several studies reported that the recurrence of angina after revascularization is particularly frequent in the first year after PCI, reaching an incidence of 32.3% [5]. The possible causes of residual angina after PCI could derive by epicardial vessels (in-stent restenosis and vasospasm), small vessels (stenosis of small side branches of residual stenosis after bifurcation treatment with one-stent and distal portion of main branch), and the microcirculation disease [6,7]. Regardless of the pathway, adding a second-line antianginal drug is suggested to control symptomatic patients in which a repeated PCI is not an option or as a temporary measure while the patient awaits a new revascularization [2].

Ranolazine is a selective inhibitor of the late sodium current (INaL) in cardiomyocytes, which is thought to be an important contributor to the pathogenesis of angina through calcium overload and an increase in oxygen consumption in the cardiomyocytes [8]. Although most studies have focused on its role in macrovascular angina, some findings suggest that ranolazine also has anti-inflammatory or antioxidant effects that may improve glycometabolic homeostasis, which are more important in microvascular angina [9]. Similarly to other antianginal drugs of recent years such as ivabradine, ranolazine has been shown to improve exercise tolerance test (ETT) parameters, reducing nitroglycerin consumption and angina frequency in people with stable angina without having any effect on blood pressure and heart rate [10,11,12]. No data are available on symptomatic patients for residual angina with certain coronary disease not suitable for PCI or bypass grafting.

We sought to evaluate the efficacy of treatment with ranolazine in addition to optimal anti-ischemic medical therapy in patients with recurrent or persistent angina following PCI with certain coronary disease of small vessels not suitable for any revascularization.

## 2. Study Population

We included 49 consecutive patients with previous coronary artery disease (CAD) already treated by PCI and stent implantation(s), who had subsequently undergone a new coronary angiography in our department for typical angina in the last three months with a Canadian Cardiovascular Society (CCS) grading of angina pectoris II or III. Patients with new epicardial coronary stenosis or in-stent restenosis were excluded; all patients presented disease of small epicardial branches (<2 mm) or with stenosis not suitable for revascularization were selected to enroll in our registry. The study complied with the Declaration of Helsinki and it was approved by local Ethics Committees (Ref.1715).

The main exclusion criteria were as follows: age <18 years; anti-ischemic drugs intolerance or hypersensitivity; epicardial coronary stenosis susceptible to revascularization; acute coronary syndrome (ACS); ejection fraction (FE) ≤ 40%;New York Heart Association (NYHA) class III to IV; CCS grading IV; persistent atrial fibrillation or flutter; pacemaker (PMK) or implantable cardioverter–defibrillator (ICD); II or III atrioventricular block; resting heart rate (HR) ≤ 50 bpm or sick sinus syndrome; left bundle branch block, left ventricular hypertrophy; aortic stenosis; mental or physical impairment leading to inability to exercise adequately; rate-corrected QT interval (QTc) greater than 500 ms or the use of drugs that prolong the QTc interval; symptomatic hypotension or uncontrolled hypertension (systolic blood pressure at rest ≥ 180 mmHg or diastolic blood pressure ≥ 100 mmHg); severe liver disease and severe renal impairment (creatinine clearance ≤ 30 mL/min); electrolyte disorders; uncontrolled thyroid disease; hemoglobin <10 g/dL; pregnancy.

### 2.1. Study Design

In this single-center, prospective, open-label, prospective registry, we evaluated the efficacy of ranolazine (375 mg twice daily) in addition to standard anti-ischemic therapy, in patients with persistent/recurrent angina after complete revascularization (as far as possible) by PCI and stent implantation (Figure 1). Patients who met the inclusion criteria, after signing the informed consent, received ranolazine at an initial dose of 375 mg twice daily, in combination with optimal medical therapy (OMT) in accordance with the most recent international guidelines [1].

The clinical, angiographic, and instrumental data (echocardiogram, ECG, EET) were collected into a database in Excel® (version 14.0; Microsoft Corporation, Redmond, USA), format. In the first phase of recruitment, the following parameters were collected: baseline clinical parameters (age, gender, height, weight, systolic and diastolic blood pressure, mean heart rate); medical history (diabetes mellitus type I or II, smoking status, hypertension, dyslipidemia, family history of cardiovascular disease, angina, previous PCI, acute myocardial infarction (AMI), coronary artery bypass grafting (CABG), chronic peripheral arterial disease (PAD), cerebral ischemia or bleeding and chronic renal failure); and echocardiographic parameters (ventricle diameters and volumes, ejection fraction, valvular disfunctions). All patients underwent clinical examination and an exercise ECG test at enrollment time (baseline: BL) and after at least 30 days (T1) from enrollment period. 

### 2.2. Exercise ECG Test (EET)

An exercise test was performed using a semirecumbent and tilting bicycle ergometer (X-SCRIBE EKG Analysis, Mortara Instruments; Ergometrics 800s, Ergoline, West-Germany) with an initial workload set at 25 Watt and increments of 25 Watt/2 min. HR and rhythm are continuously recorded using a 12-lead electrocardiogram; blood pressure was measured at baseline, at peak exercise, and during the last minute of each stage, including recovery. Exercise-induced ECG ischemia was defined as the new development of horizontal or down-sloping ST-segment depression (≥1 mm at 80 milliseconds after the J point) versus baseline tracing. For patients with permitted baseline ST depression at rest (<1 mm), qualifying ST segment depression is defined as additional ST depression of at least 1 mm below the resting value. In the EET, we evaluated the presence and time of occurrence of angina or ST segment deviations, the total exercise duration, the maximum double product, the occurrence of arrhythmias, systolic blood pressure (SBP) and heart rate (HR) at peak and at rest, finally we collected the workload (expressed in watts). Each test was preceded by physical examination, 12-lead ECG at rest, and questions to verify adherence to therapy in the past 30 days (T1). The stress test was interrupted if the patient developed chest pain or a significant adverse event (significant ventricular arrhythmia, limiting breathlessness, dizziness, muscular exhaustion, chest pain, arterial pressure drop ≥ 10 mmHg with symptoms, or severe systemic hypertension).

### 2.3. Endpoint

We sought to evaluate the effects of ranolazine, after 30 days, in association with OMT comparing the effects to baseline before the study drug administration, in terms of symptoms (exercise angina or anginal equivalent), ECG signs of myocardial ischemia, and exercise duration after 30 days of treatment (T1), evaluated by EET. We chose a 1-month follow-up to investigate the effect of ranolazine for a tight symptom control and for a rapid improvement of the clinical status, considering the exercise parameters. 

### 2.4. Statistical Analysis

Continuous variables were presented as mean ± standard deviation (SD) and categorical variables were presented as percentages. Categorical variables were compared among groups using the chi-square test or Fisher’s exact test when appropriate, whereas continuous variables were compared with Student’s *t*-test. All tests were two-sided, and a *p*-value less than 0.05 was considered statistically significant. All analyses were performed using SPSS software version 20.0 (IBM Corporation, Armonk, NY, USA).

## 3. Results

Forty-nine consecutive patients with persistent or recurrent stable angina after PCI were included in our prospective registry. The baseline clinical characteristics are illustrated in Table 1. The average age of the population was 62.6 ± 11.3 years old; 44 patients (89.8%) were male. The cardiovascular risk factors were largely represented; especially, about half of the study population reported a previous myocardial infarction, and 65% were smokers. Table 1 also shows the basic echocardiographic features at the baseline. Baseline medical therapy is shown in Table 2; particularly, dual antiplatelet therapy was administrated in almost all patients (94%) with clopidogrel in 59.2%, prasugrel in 26.5%, and ticagrelor in 8.1%; the nitrates were widely used. EET was performed in all patients at baseline and at 30 days (T1) after enrollment (Table 3). Comparing the EET data before and after the therapy with ranolazine, the results at 30 days were improved. The duration of the exercise was longer at 30 days (T1 9’10” ± 2’ versus BL 8’10” ± 2’, *p* = 0.01), as shown in Figure 2. 

Moreover, at T1, the patients had highlighted a greater resistance exercise compared to BL; only 7 patients (14.3%) at T1 had not exceeded the threshold of six minutes (75 watts) compared to 20 (40.8%) at BL (*p* = 0.0003). Stress angina was lesser reported after 1 month with significant differences in all tests along (T1 4.1% versus BL 16.3%, *p* = 0.04) and before the third step, even if with only a positive trend in favor of ranolazine (Figure 3). Furthermore, the exercise-induced arrhythmias were reduced after the adding of ranolazine (BL 30.6% versus T1 14.3%, *p* = 0.05). No other effects were recorded on heart rate and systemic blood pressure. A normal increase of HR and BP was observed during the stress test if comparing with the hypothetic effects of a high dose of beta-blockers. 

## 4. Discussion

Our preliminary results showed that the combination of low doses of ranolazine (375 mg twice daily) to optimal medical therapy (OMT), in patients with persistent or recurrent stable angina after PCI, can reduce the incidence of symptoms and increase exercise tolerance during EET. At one-month follow-up, the incidence of exercise-induced arrhythmias was lower.

In the RIVER-PCI (Ranolazine for Incomplete Vessel Revascularization Post-Percutaneous Coronary Intervention) trial, ranolazine did not reduce the composite of ischaemia-driven revascularization or hospitalization without revascularization in 2651 patients with a history of chronic angina and incomplete revascularization after PCI, including those with and without PCI for a CAD indication, nor did it reduce angina symptoms at 1 year [13,14]. These results support the use of ranolazine only as a second-line drug in CCS patients with refractory angina despite commonly used antianginal agents such as beta-blockers, CCBs, and/or long-acting nitrates (Class IIa–2019 ESC Guidelines for the diagnosis and management of chronic coronary syndromes) [2].

RIVER-PCI trial investigators defined incomplete revascularization as the presence of at least one lesion with stenosis of 50% or more in diameter (visually estimated) in a coronary artery with a reference vessel diameter of 2 mm or more, whether in a percutaneous coronary intervention-treated target vessel or in a non-treated, non-target vessel [14]. A 2015 meta-analysis by Belsey et al. (that included 46 studies) suggested that several second-line anti-ischaemic drugs (such as long-acting nitrates, ranolazine, trimetazidine, and ivabradine) can be useful in combination with first-line therapy (beta-blockers and CCB) [15].

Therefore, there is a lack of evidence to support the use of ranolazine in patients with CCS following PCI with incomplete revascularization of epicardial vessels suitable for additional PCI. However, in the current scenario, it is probably more important to understand if there is a role of this drug in residual ischemia after PCI not suitable for further revascularization. 

Our study addressed the problem of patients with complete revascularization (as far as possible) but with small epicardial vessels disease (<2 mm). This happens very often during the treatment of bifurcations with small lateral branches, or in the disease of distal vessels [16]. Often, these branches do not give significant or lasting symptoms over time; however, in some cases, the stenosis of these branches results in persistent stable angina [17]. Our intent was to directly quantify the benefit of ranolazine in terms of functional capacity, but the sample size does not allow reliable data on major cardiovascular events during follow-up. On the other hand, the design of our study may allow the clinician to have a more direct and concrete idea of the effects of the drug he prescribes, with consequent impact on the daily life of patients. The measurement of these effects in terms of double product (DP), exercise resistance (Watt), and duration of the exercise has different purposes than verifying the clinical outcome, for which we refer to the great trials on ranolazine.

In order to enhance the practical implication of our results, we arbitrarily focused our attention on the third step of the EET, when after two minutes at the 75 watt load, the resistance increases to 100 watts (6 min).

Looking at the literature, the workload imposed by the resistance of 75 watts on a man with an average weight of 70 kg corresponds to the efforts that most people face in everyday life [18]. Shoveling the snow requires approximately 5.1 metabolic equivalents (METS), which corresponds to 89 watts, and a similar effort is required to cut the wood. In real life, there are many other variables to consider, such as emotional factors and ambient temperature. However, it is very interesting to note that our results show the ability of ranolazine to guarantee the achievement of the 100-watt threshold more often than OMT alone.

A critical aspect in the discussion of our study is the certainty that angina symptoms are related to small caliber branches. Indeed, the persistence or recurrence of angina after successful and complete PCI affects from one-fifth to one-third of patients undergoing revascularization at one-year follow-up. There are multiple underlying causes. Restenosis or coronary atherosclerosis progression explain symptom recurrence in some patients, while functional causes, including vasomotor abnormalities of epicardial coronary arteries and/or coronary microvascular dysfunction, could explain symptoms in the other patients [19]. In this case, our results could also be explained thanks to other pharmacodynamic mechanisms that overlap with the anti-ischemic mechanism on diseased epicardial vessels. In recent years, some studies showed how ranolazine may be an effective treatment option also for the symptomatic management of microvascular angina [20,21,22,23].

The limitations of our trial should be considered. First, the sample size of the present study was too small to test clinical endpoints. Along this line, the study was certainly not powered to evaluate adverse clinical events. Second, the borderline statistical differences in other outcome measures, although significant, were driven by sample size. Although our results have suggested a possible improvement of exercise duration, the reduction of angina symptoms, and arrythmias during the stress test, this observational study is limited by selection bias and confounding. Trying to reduce possible bias, we observed the same patients before and after therapy. This should be kept in mind when interpreting the study findings.

## 5. Conclusions

In patients with persistent/recurrent angina and residual ischemia due to small epicardial arterial branches not suitable for further revascularization after PCI, the addition of a low dose of ranolazine (375 mg b.i.d.) to standard anti-ischemic therapy showed significant improvement in stress EET results after one month of therapy, including reduced exercise angina, increased exercise tolerance and duration, and reduced exercise arrhythmias.

## Figures and Tables

**Figure 1 jcm-09-02110-f001:**
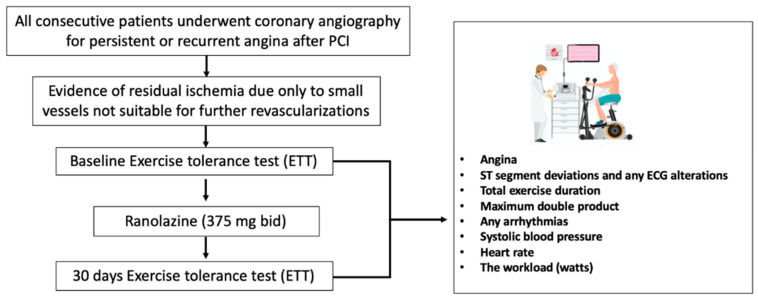
The figure illustrates the study design and study population (PCI: percutaneous coronary intervention).

**Figure 2 jcm-09-02110-f002:**
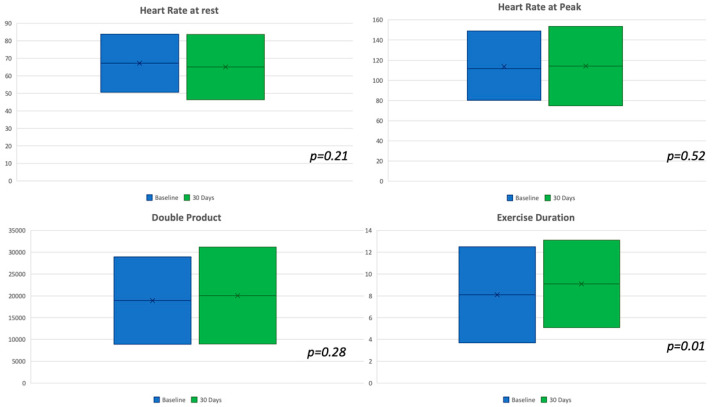
Graphical representations for the continuous variables at baseline and at follow-up show a significant difference in exercise duration that was increased at follow-up.

**Figure 3 jcm-09-02110-f003:**
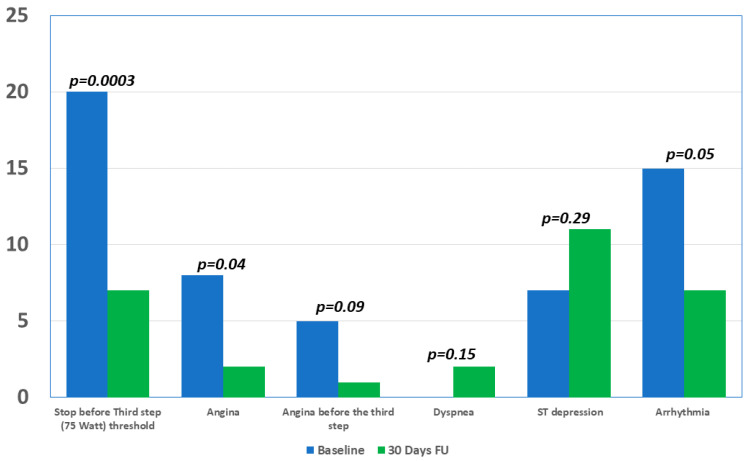
Graphical representation of results at baseline and at follow-up, showing a lesser rate of angina symptoms, arrhythmia, and an increase workload in terms of watts.

**Table 1 jcm-09-02110-t001:** Baseline clinical and echocardiographic features of the study population.

	Population (*N* = 49)
**Age (years) ± SD**	62.6 ± 11.2
**Gender (male)**	44 (89.8%)
**Height (cm) ± SD**	172.2 ± 7.02
**Previous AMI**	25 (51.1%)
**Previous CABG**	18 (36.7%)
**Hypertension**	29 (59.2%)
**Smoker**	32 (65.3%)
**Familiarity**	22 (44.8%)
**Dyslipidaemia**	32 (65.3%)
**Diabetes Mellitus II**	9 (18.3%)
**Stroke/TIA**	4 (8.1%)
**Chronic Renal failure**	3 (6.1%)
**Ejection Fraction (%) ± SD**	48.3 ± 5.7
**WMSI ± dS**	1.3 ± 0.39
**End-diastolic diameter (mm) ± SD**	51.7 ± 5.2
**End-systolic diameter (mm) ± SD**	37 ± 3.6

AMI: acute myocardial infarction. CABG: coronary artery bypass grafting. TIA: transient ischemic attack. WMSCI: wall motion score index.

**Table 2 jcm-09-02110-t002:** Medical therapy at baseline (BL).

	Population (*N* = 49)
**Acetylsalicylic acid (ASA)**	47 (95.9%)
**Clopidogrel**	29 (59.2%)
**Prasugrel**	13 (26.5%)
**Ticagrelor**	4 (8.1%)
**Beta-blockers**	41 (83.7%)
**Nitrates**	33 (67.3%)
**Calcium channel blockers**	4 (8.1%)
**ACE inhibitors**	41 (83.7%)
**Angiotensin receptor Antagonists**	9 (18.3%)
**Statins**	44 (89.8%)

**Table 3 jcm-09-02110-t003:** Results of the Exercise echocardiogram (ECG) Test (EET) at baseline (BL) and after 30 days of therapy (T1). (SBP: systolic blood pressure. HR: heart rate).

	BL (*N* = 49)	T1 (*N* = 49)	Mean Difference (CI 95%)	*p*
**Heart rate at rest (bpm) ± SD**	67.2 ± 8.3	65.0 ± 9.3	2.2 (CI: −1.33 to 5.73)	0.21
**Heart rate at peak (bpm) ± SD**	111.7 ± 18.7	114.2 ± 19.7	−2.5 (CI −10.20 to 5.20)	0.52
**Systolic blood pressure at rest (mmHg) ± SD**	119.5 ± 15.2	125.9 ± 14.9	−6.4 (CI: −12.44 to −0.36)	0.03
**Diastolic blood pressure at rest (mmHg) ± SD**	78.5 ± 14.2	75.6 ± 8.9	2.9 (CI: −1.85 to 7.65)	0.22
**Systolic blood pressure at peak (mmHg) ± SD**	167.9 ± 23.9	176.4 ± 22.7	−8.5 (CI: −56.54 to 39.54)	0.07
**Diastolic blood pressure at peak (mmHg) ± SD**	94.3 ± 21.6	90.4 ± 22.7	3.9 (CI: −4.98 to 12.78)	0.38
**Double product (SBP x HR at peak)**	18,913 ± 5008.4	20,066 ± 5548.9	−1153 (CI: −3272.6 to 966.6)	0.28
**Exercise duration (min) ± SD**	8’1” ± 2’2”	9’1” ± 2’0”	1’ (CI: −1.84 to −0.15)	0.01
			**Odds Ratio (CI 95%)**	
**Stop before third step (75 Watt)** **threshold**	20 (40.8%)	7 (14.3%)	0.24 (CI: 0.09 to 0.64)	0.0003
**Angina**	8 (16.3 %)	2 (4.1%)	0.21 (CI: 0.043 to 1.08)	0.04
**Angina before the third step**	5 (10.2%)	1 (2.0%)	0.18 (CI: 0.0206 to 1.63)	0.09
**Dyspnea**	0 (0.0%)	2 (4.1%)		0.15
**ST depression**	7 (14.3%)	11 (22.4%)	1.73 (CI: 0.6113 to 4.93)	0.29
**Arrhythmia**	15 (30.6%)	7 (14.3%)	0.37 (CI: 0.1383 to 1.03)	0.05

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
