# Peer review of "The Role of Ranolazine for the Treatment of Residual Angina beyond the Percutaneous Coronary Revascularization"

_jcm, 2020, doi:10.3390/jcm9072110_

Round 1
Reviewer 1 Report
Major comments:
Unfortunately the chosen outcome measures are not clinical end points and thereby of less interest for clinicians. Beyond the missing clinical end points: Though the presented difference regarding average duration of the exercise may statistically be significant, however, it is rather meaningless clinically (RG 9'1'' ± 2' vs. BL 8'1'' ± 2', p=0.01). Furthermore, differences in several other outcome measures exhibit only marginal or no statistical significance (stress angina: p=0.04, exercise-induced arrhythmias: p=0.05) showing lack of statistical power. Therefore, the conclusions drawn by the authors are not fully supported by the presented data.
Minor comments:
Part of Figure 1 with the schematized exercise test is inappropriate for a scientific journal and should be omitted.
Wording is occasionally unusual e.g., angor instead of angina.
Author Response
Major comments:
Unfortunately the chosen outcome measures are not clinical end points and thereby of less interest for clinicians. Beyond the missing clinical end points: Though the presented difference regarding average duration of the exercise may statistically be significant, however, it is rather meaningless clinically (RG 9'1'' ± 2' vs. BL 8'1'' ± 2', p=0.01). Furthermore, differences in several other outcome measures exhibit only marginal or no statistical significance (stress angina: p=0.04, exercise-induced arrhythmias: p=0.05) showing lack of statistical power. Therefore, the conclusions drawn by the authors are not fully supported by the presented data.
- Thank you for your careful revision, your contribution it is important to improve our paper. We fully agree with you about the endpoint choice, we wanted to test the ranolazine effect on exercise parameters. The marginal statistical significances are correlated to the small sample-size. To address your observations, we added the limits paragraph in the paper (page 9).
Minor comments:
Part of Figure 1 with the schematized exercise test is inappropriate for a scientific journal and should be omitted.
- The two reviewers do not agree about the Figure. We leave to the editor the final decision to omitted or confirm the figure.
Wording is occasionally unusual e.g., angor instead of angina.
- Thank you, we changed “angor” with “angina” in the text and we revised all unusual words.
To address your revisions and suggestions, we improved the design description, we better described the methods, results and conclusions paragraphs. Furthermore, we checked all paper improving the English language and style.
Reviewer 2 Report
I read with interest the study entitled "The role of Ranolazine for the treatment of residual angina beyond the percutaneous coronary revascularization" by Calcagno et al. The authors evaluated the role of ranolazine in patients with persistent angina following PCI. My comments are as follows:
- Abstract: replace "optical" with "optimal"
- Table "4" should be replaced by "3".
- The authors should provide in Table 3 the mean difference (95%CI).
- Graphical representation of the provided information is encouraged. i.e. box-plot for the continuous variables at baseline and at follow-up and histograms for the binary outcomes.
- The authors should comment on potential confounding
- The authors should explain why they chose the 30 days follow-up
- The authors should provide details on the abbreviations in each Table.
Author Response
Abstract: replace "optical" with "optimal"
- Thank you, we changed it in the paper.
Table "4" should be replaced by "3".
- Thank you, we changed it in the paper.
The authors should provide in Table 3 the mean difference (95%CI).
- Thanks for the important suggestion. We added the mean difference with relative confidence interval. For the percentage values (categorical variables) ​​we added odds ratio with a confidence interval. (See Tab 3).
Graphical representation of the provided information is encouraged. i.e. box-plot for the continuous variables at baseline and at follow-up and histograms for the binary outcomes.
- Thanks for the suggestion. We added histograms for the graphical representation of binary outcomes and box-plot (with mean and CI) for the continuous variables at baseline and at follow-up.
The authors should comment on potential confounding
- Thank you, we agree with you. We added a comment on potential confounding in limits paragraph.
The authors should explain why they chose the 30 days follow-up
- Thank you for your observation. We chose a 30 days follow-up because we wanted to test the effect of ranolazine with for a tight symptom control and rapid improvement of the clinical status, considering the exercise parameters. In our opinion, these reflected a potential increasing of the quality of life. We added this clarification in the text (Endpoint paragraph page 6).
The authors should provide details on the abbreviations in each Table.
- Thank you, we explained all abbreviation for each table in the relative caption/legend.
To address your revisions and suggestions, we improved the design description, we better described the methods, results and conclusions paragraphs. Furthermore, we checked all paper improving the English language and style.
Round 2
Reviewer 2 Report
The authors have addressed my comments.